# IDF++: Analyzing and Improving Integer Discrete Flows for Lossless Compression

**Rianne van den Berg, Alexey A. Gritsenko, Mostafa Dehghani,**
**Casper Kaae Sønderby & Tim Salimans**
Google Research
`{riannevdberg,agritsenko,dehghani,casperkaae,salimans}@google.com`

## Abstract

In this paper we analyse and improve integer discrete flows for lossless compression. Integer discrete flows are a recently proposed class of models that learn invertible transformations for integer-valued random variables. Their discrete nature makes them particularly suitable for lossless compression with entropy coding schemes. We start by investigating a recent theoretical claim that states that invertible flows for discrete random variables are less flexible than their continuous counterparts. We demonstrate with a proof that this claim does not hold for integer discrete flows due to the embedding of data with finite support into the countably infinite integer lattice. Furthermore, we zoom in on the effect of gradient bias due to the straight-through estimator in integer discrete flows, and demonstrate that its influence is highly dependent on architecture choices and less prominent than previously thought. Finally, we show how different architecture modifications improve the performance of this model class for lossless compression, and that they also enable more efficient compression: a model with half the number of flow layers performs on par with or better than the original integer discrete flow model.

## 1 Introduction

Density estimation algorithms that minimize the cross entropy between a data distribution and a model distribution can be interpreted as lossless compression algorithms because the cross-entropy upper bounds the data entropy. While autoregressive neural networks (Uria et al., 2014; Theis & Bethge, 2015; Oord et al., 2016; Salimans et al., 2017) and variational auto-encoders (Kingma & Welling, 2013; Rezende & Mohamed, 2015) have seen practical connections to lossless compression for some time, normalizing flows were only recently used for lossless compression. Most normalizing flow models are designed for real-valued data, which complicates an efficient connection with entropy coders for lossless compression since entropy coders require discretized data. However, normalizing flows for real-valued data were recently connected to bits-back coding by Ho et al. (2019b), opening up the possibility for efficient dataset compression with high compression rates. Orthogonal to this, Tran et al. (2019) and Hoogeboom et al. (2019a) introduced normalizing flows for discrete random variables. Hoogeboom et al. (2019a) demonstrated that integer discrete flows can be connected directly to entropy coders without the need for bits-back coding.

In this paper we aim to improve integer discrete flows for lossless compression. Recent literature has proposed several hypotheses on the weaknesses of this model class, which we investigate as potential directions for improving compression performance. More specifically, we start by discussing the claim on the flexibility of normalizing flows for discrete random variables by Papamakarios et al. (2019), and we show that this limitation on flexibility does not apply to integer discrete flows. We then continue by discussing the potential influence of gradient bias on the training of integer discrete flows. We demonstrate that other less-biased gradient estimators do not improve final results. Furthermore, through a numerical analysis on a toy example we show that the straight-through gradient estimates for 8-bit data correlate well with finite difference estimates of the gradient. We also demonstrate that the previously observed performance degradation as a function of number of flows is highly dependent on the architecture of the coupling layers. Motivated by this last finding, we introduce several architecture changes that improve the performance of this model class on lossless image compression.

## 2 RELATED WORK

**Continuous Generative Models:** Continuous generative flow-based models (Chen & Gopinath, 2001; Dinh et al., 2014; 2016) are attractive due to their tractable likelihood computation. Recently these models have demonstrated promising performance in modeling images (Ho et al., 2019a; Kingma & Dhariwal, 2018), audio (Kim et al., 2018), and video (Kumar et al., 2019). We refer to Papamakarios et al. (2019) for a recent comprehensive review of the field.

By discretizing the continuous latent vectors of variational auto-encoders and flow-based models, efficient lossless compression can be achieved using bits-back coding (Hinton & Van Camp, 1993). Recent examples of such approaches are Local Bits-Back Coding with normalizing flows (Ho et al., 2019b) and variational auto-encoders with bits-back coding such as Bits-Back with ANS (Townsend et al., 2019b), Bit-Swap (Kingma et al., 2019) and HiLLoC (Townsend et al., 2019a). These methods achieve good performance when compressing a full dataset, such as the ImageNet test set, since the auxiliary bits needed for bits-back coding can be amortized across many samples. However, encoding a single image would require more bits than the original image itself (Ho et al., 2019b).

**Learned discrete lossless compression:** Producing discrete codes allows entropy coders to be directly applied to single data instances. Mentzer et al. (2019) encode an image into a set of discrete multiscale latent vectors that can be stored efficiently. Fully autoregressive generative models condition unseen pixels directly on the previously observed pixel values and have achieved the best likelihood values compared to other models (Oord et al., 2016; Salimans et al., 2017). However, decoding with these models is impractically slow since the conditional distribution for each pixel has to be computed sequentially. Recently, super-resolution networks were used for lossless compression (Cao et al., 2020) by storing a low resolution image in raw format and by encoding the corrections needed for lossless up-sampling to the full image resolution with a partial autoregressive model. Finally, Mentzer et al. (2020) first encode an image using an efficient lossy compression algorithm and store the residual using a generative model conditioned on the lossy image encoding.

**Hand-designed Lossless Compression Codecs:** The popular PNG algorithm (Boutell & Lane, 1997) leverages a simple autoregressive model and the DEFLATE algorithm (Deutsch, 1996) for compression. WebP (Rabbat, 2010) uses larger patches for conditional compression coupled with a custom entropy coder. In its lossless mode, JPEG 2000 (Rabbani, 2002) transforms an image using wavelet transforms at multiple scales before encoding. Lastly, FLIF (Sneyers & Wuille, 2016) uses an adaptive entropy coder that selects the local context model using a per-image learned decision tree.

## 3 BACKGROUND: NORMALIZING FLOWS

In this section we briefly review normalizing flows for real-valued and discrete random variables. A normalizing flow consists of a sequence of invertible functions applied to a random variable $\boldsymbol{x}$: $f^K \circ f^{K-1} \circ ... \circ f^1(\boldsymbol{x})$, yielding random variables $\boldsymbol{y}^K \leftarrow ... \leftarrow \boldsymbol{y}^1 \leftarrow \boldsymbol{y}^0 = \boldsymbol{x}$. First, consider a real-valued random variable $\boldsymbol{x} \in \mathbb{R}^d$ with unknown distribution $p_{\boldsymbol{x}}(\boldsymbol{x})$. Let $f : \mathbb{R}^d \mapsto \mathbb{R}^d$ be an invertible function that such that $\boldsymbol{y} = f(\boldsymbol{x})$ with $\boldsymbol{y} \in \mathbb{R}^d$. If we impose a density $p_{\boldsymbol{y}}(\boldsymbol{y})$ on $\boldsymbol{y}$, the distribution $p_{\boldsymbol{x}}(\boldsymbol{x})$ is obtained by marginalizing out $\boldsymbol{y}$ from the joint distribution $p_{\boldsymbol{x},\boldsymbol{y}}(\boldsymbol{x},\boldsymbol{y}) = p_{\boldsymbol{x}|\boldsymbol{y}}(\boldsymbol{x}|\boldsymbol{y})p_{\boldsymbol{y}}(\boldsymbol{y})$:

$$p_{\boldsymbol{x}}(\boldsymbol{x}) = \int \delta(\boldsymbol{x} - f^{-1}(\boldsymbol{y})) p_{\boldsymbol{y}}(\boldsymbol{y}) \mathrm{d}\boldsymbol{y} = \int \delta(\boldsymbol{x} - \boldsymbol{u}) p_{\boldsymbol{y}}(f(\boldsymbol{u})) \left| \det \frac{\partial f(\boldsymbol{u})}{\partial \boldsymbol{u}} \right| \mathrm{d}\boldsymbol{u} = p_{\boldsymbol{y}}(f(\boldsymbol{x})) \left| \det \frac{\partial f(\boldsymbol{x})}{\partial \boldsymbol{x}} \right|, \tag{1}$$

where we used $p_{\boldsymbol{x}|\boldsymbol{y}}(\boldsymbol{x}|\boldsymbol{y}) = \delta(\boldsymbol{x} - f^{-1}(\boldsymbol{y}))$, with $\delta(x - x')$ the Dirac delta distribution, and we applied a change of variables. Repeated application of (1) for a sequence of transformations then yields the log-probability:

$$\ln p_{\boldsymbol{x}}(\boldsymbol{x}) = \ln p_{\boldsymbol{y}^K}(\boldsymbol{y}^K) + \sum_{k=1}^{K} \ln \left| \det \frac{\partial \boldsymbol{y}^k}{\partial \boldsymbol{y}^{k-1}} \right|. \tag{2}$$

By parameterizing the invertible functions with invertible neural networks and by choosing a tractable distribution $p_{\boldsymbol{y}^K}(\boldsymbol{y}^K)$ these models can be used to optimize the log-likelihood of $\boldsymbol{x}$. When modeling discrete data with continuous flow models, dequantization noise must be added to the input data to ensure that a lower bound to the discrete log-likelihood is optimized (Uria et al., 2013; Theis et al., 2015; Ho et al., 2019a).

### 3.1 DISCRETE NORMALIZING FLOWS

Next, consider $\boldsymbol{x}$ to be a discrete random variable with domain $\mathcal{X}$, and define the invertible function $f : \mathcal{X} \mapsto \mathcal{X}$. In general $f$ can be a mapping between two different domains, but for our discussions it will be sufficient to consider a single domain. The marginal probability mass of $\boldsymbol{x}$ is given by

$$p_{\boldsymbol{x}}(\boldsymbol{x}) = \sum_{\boldsymbol{y} \in \mathcal{X}} p_{\boldsymbol{x}|\boldsymbol{y}}(\boldsymbol{x}|\boldsymbol{y}) p_{\boldsymbol{y}}(\boldsymbol{y}) = \sum_{\boldsymbol{y} \in \mathcal{X}} \delta_{\boldsymbol{x}, f^{-1}(\boldsymbol{y})} p_{\boldsymbol{y}}(\boldsymbol{y}) = p_{\boldsymbol{y}}(f(\boldsymbol{x})) , \tag{3}$$

with the Kronecker delta function $\delta_{i,j} = 1$ if $i = j$ and $0$ otherwise. Note the absence of a volume correction in the form of a Jacobian determinant owing to the fact that probability mass functions only have support on a discrete set of points.

Recently, Tran et al. (2019) and Hoogeboom et al. (2019a) have both considered normalizing flows for discrete random variables. In integer discrete flows (IDF) by Hoogeboom et al. (2019a) the random variables are assumed to be integers, i.e. $\mathcal{X} = \mathbb{Z}^d$. The main building block of IDF is an additive bipartite coupling layer (Dinh et al., 2016):

$$\begin{bmatrix} \boldsymbol{y}_a \\ \boldsymbol{y}_b \end{bmatrix} = \begin{bmatrix} \boldsymbol{x}_a \\ \boldsymbol{x}_b + \lfloor \boldsymbol{t}_{\boldsymbol{\theta}}(\boldsymbol{x}_a) \rceil \end{bmatrix} . \tag{4}$$

Here $\boldsymbol{y}_a \in \mathbb{Z}^m$, $\boldsymbol{y}_b \in \mathbb{Z}^n$ are obtained by splitting $\boldsymbol{y} \in \mathbb{Z}^d$ into two pieces such that $m + n = d$, and similarly for $\boldsymbol{x}_a$ and $\boldsymbol{x}_b$. The pre-quantized translation $\boldsymbol{t}_{\boldsymbol{\theta}}(\cdot)$ is represented by the output of a neural network with learnable parameters $\boldsymbol{\theta}$, and is rounded to integer values with the rounding operator $\lfloor \cdot \rceil$. The parameters $\boldsymbol{\theta}$ are optimized with a straight-through estimator and a gradient-based optimizer.

Tran et al. (2019) introduce flows for non-ordinal discrete random values with a finite number of possible values: $\boldsymbol{x} \in \mathcal{X} = \{0, 1, ..., K - 1\}^d$. They introduce an autoregressive and a coupling bijector, with the coupling layer given by $[\boldsymbol{y}_a, \boldsymbol{y}_b] = [\boldsymbol{x}_a, (\boldsymbol{s}_{\boldsymbol{\theta}}(\boldsymbol{x}_a) \circ \boldsymbol{x}_b + \boldsymbol{t}_{\boldsymbol{\theta}}(\boldsymbol{x}_a)) \bmod K]$, with $\circ$ denoting element-wise multiplication and $\boldsymbol{s} \in \{1, 2..., K - 1\}$ and $\boldsymbol{t} \in \{0, 1, ..., K - 1\}$ and $\boldsymbol{s}$ and $K$ constrained to be co-prime. Gradients are again computed with a straight-through estimator.

One of the main differences between discrete flows (Tran et al., 2019) and integer discrete flows (Hoogeboom et al., 2019a) is that the former treats the random variable as having a finite number of classes while the latter considers a countably infinite number of classes. In the next section we will see that this difference plays an important role in the theoretical flexibility of invertible flows for discrete random variables.

## 4 CAN INTEGER DISCRETE FLOWS FACTORIZE ANY DISTRIBUTION?

In this section we discuss the theoretical flexibility of normalizing flows for discrete random variables. In particular, we will focus on the ability to map from a distribution with dependencies across all dimensions to a distribution which is fully factorized (i.e. independent across all dimensions). As generative flow models require a tractable base distribution with efficient sampling, simple base distributions (e.g. Gaussian, logistic, categorical) which are independent across dimensions/subpixels are frequently used. Papamakarios et al. (2019) state that invertible flows for discrete random variables cannot map all distributions to a factorized base distribution, and name this as a limitation as compared to flows for real-valued random variables. We will analyze this claim and show that this limitation can be overcome by embedding the discrete data into a space with a larger set of possible values. Since integer discrete flows embed the data into the integer lattice $\mathbb{Z}^d$, this model class naturally does not suffer from a limited factorization capacity.

The starting point for this discussion lies in the observation by Papamakarios et al. (2019) that invertible normalizing flows for discrete random variables can only permute probability masses in the probability tensor that represents the probability distribution of the random variable. In other words, if we have an invertible function $f : \mathcal{X} \mapsto \mathcal{X}$, then there is always exactly one $\boldsymbol{y}$, such that $\boldsymbol{y} = f(\boldsymbol{x})$ and $p_{\boldsymbol{x}}(\boldsymbol{x}) = p_{\boldsymbol{y}}(\boldsymbol{y})$. In contrast, non-volume preserving normalizing flows for real-valued random variables can increase or decrease the density through the Jacobian determinant in the change of variables formula in (1).

Papamakarios et al. (2019) then discuss an educative example to show that this permutation property can lead to a limited ability to factorize distributions. Consider the case of a two-dimensional random

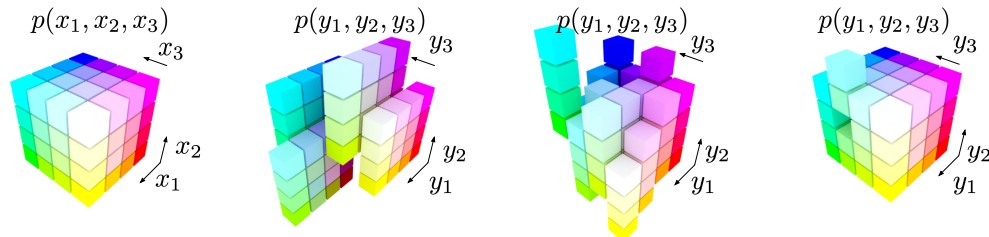

Figure 1: Left: 3D probability distribution tensor, only nonzero values are indicated with colored cubes, all empty space is assumed to be filled with zero-valued cubes. Middle left: an example of an additive transformation conditioned on $x_3$: $y_1 = x_1 + \lfloor t_1(x_3) \rceil$, $y_2 = x_2 + \lfloor t_2(x_3) \rceil$, $y_3 = x_3$. Middle right: an example of an additive transformation conditioned on $x_1$ and $x_3$: $y_1 = x_1$, $y_2 = x_2 + \lfloor t_2(x_1, x_3) \rceil$, $y_3 = x_3$. Right: a distribution tensor that a single additive transformation of the form of (4) cannot generate from the cube on the left.

variable $\boldsymbol{x} = (x_1, x_2)$, with $x_1, x_2 \in \{0, 1\}$, and a data distribution given by

$$p_{\boldsymbol{x}}(x_1, x_2): \quad \begin{array}{c c} x_1 \backslash x_2 & \begin{array}{cc} 0 & 1 \end{array} \\ \begin{array}{c} 0 \\ 1 \end{array} & \begin{pmatrix} 0.1 & 0.3 \\ 0.2 & 0.4 \end{pmatrix} \end{array}. \tag{5}$$

In order to map this probability distribution to an independent base distribution, the corresponding probability matrix must be of rank one. In other words, given a sequence of functions $f^k : \mathcal{X} \mapsto \mathcal{X}$ with $\mathcal{X} = \{0, 1\}^2$, the probability distribution of the random variable $\boldsymbol{y} = \boldsymbol{y}^K = f^K \circ f^{K-1} \circ ... \circ f^1(\boldsymbol{x})$ should be represented by a matrix with linearly dependent columns or rows. This allows for a factorization into an outer product of two vectors that represent the independent base distributions for $y_1$ and $y_2$. Since discrete flows can only permute probability mass tensors, the matrix corresponding to $p_{\boldsymbol{y}}(y_1, y_2)$ must be a permutation of the matrix in (5). However, there is no permutation of the elements in (5) that results in a rank-one matrix. Therefore, Papamakarios et al. (2019) conclude that discrete normalizing flows cannot map any distribution to a factorized base distribution.

However, one of the key assumptions made above is that the domain of $f^k$ is restricted to $\mathcal{X} = \{0, 1\}^2$. By extending $\mathcal{X}$ to a larger number of classes this example *can* in fact be factorized by a discrete normalizing flow. More concretely, let us extend the domain to $\mathcal{X} = \{0, 1, 2, 3\}^2$. The probability distribution matrix of $\boldsymbol{x}$ is shown below, together with a permutation of rank 1.

$$p_{\boldsymbol{x}}(x_1, x_2): \begin{pmatrix} 0.1 & 0.3 & 0 & 0 \\ 0.2 & 0.4 & 0 & 0 \\ 0 & 0 & 0 & 0 \\ 0 & 0 & 0 & 0 \end{pmatrix} \rightarrow p_{\boldsymbol{y}}(y_1, y_2): \begin{pmatrix} 0.1 & 0 & 0 & 0 \\ 0.2 & 0 & 0 & 0 \\ 0.3 & 0 & 0 & 0 \\ 0.4 & 0 & 0 & 0 \end{pmatrix} = \begin{pmatrix} 0.1 \\ 0.2 \\ 0.3 \\ 0.4 \end{pmatrix} \otimes \begin{pmatrix} 1 \\ 0 \\ 0 \\ 0 \end{pmatrix}. \tag{6}$$

This example illustrates that the number of classes that are considered valid for the discrete random variables plays a crucial role in the flexibility of discrete flows. Here, we claim that this holds more generally through the following two lemmas. First we show that by embedding the data into a space with more possible values than present in the data itself, one can always construct an invertible mapping to one-dimensional variable embedded in $d$ dimensions, and that the resulting variable has a distribution which is trivially factorized across all $d$ dimensions.

**Lemma 1.** *A $d$-dimensional discrete random variable $\boldsymbol{x} = (x_1, \ldots, x_d)^T$ with $d > 1$ and $x_i \in \{0, \ldots, K^{(i)} - 1\}$, distributed according to an arbitrary distribution $p_{\boldsymbol{x}}(\boldsymbol{x})$, can be transformed to a one-dimensional random variable $y \in \{0, \ldots, \left(\prod_{i=1}^d K^{(i)}\right) - 1\}$ with a number of classes that scales exponentially with the dimension $d$ through a bijective mapping $f$. Embedding $y$ in $d$ dimensions through padding with $d - 1$ zeros, i.e. $\boldsymbol{y} = (y, 0, \ldots, 0)^T$, the distribution over $\boldsymbol{y}$ is trivially factorized across all dimensions: $p_{\boldsymbol{x}}(\boldsymbol{x}) = p_y(f(\boldsymbol{x})) \prod_{i=2}^d \delta_{y_i, 0}$ with $\delta_{i,j}$ the Kronecker delta distribution for all zero-padded dimensions.*

The closed form of the mapping is $y = f(\boldsymbol{x}) = x_1 + K^{(1)} x_2 + K^{(1)} K^{(2)} x_3 + \cdots + \prod_{i=1}^{d-1} K^{(i)} x_d$, such that $y \in \{0, \ldots, \left(\prod_{i=1}^d K^{(i)}\right) - 1\}$. See Appendix A for the full proof. Intuitively, given enough classes, a factorized base distribution is obtained by "flattening" the hypercube that contains

Figure 2: Visualization of an IDF that has learned to factorize the probability distribution of the toy example in (5). Left: empirical densities of the data, the data transformed by one additive bijector, and the data transformed by two additive bijectors. Right: similar to the left plot, but with data sampled from the model.

all nonzero entries in the data distribution tensor (see left panel Figure 1) into one dimension. The following lemma demonstrates that additive coupling layers as in (4) of integer discrete flows can model such an invertible mapping.

**Lemma 2.** *For a $d$-dimensional random variable $x$, translation operations of the form $z_a = x_a$, $z_b = x_b + \lfloor t(x_a) \rceil$, with $a$ and $b$ denoting the indices of two splits of the variable $x$, are sufficient to map $x$ in an invertible manner to a random variable $y = (y_1, 0, \ldots, 0)^T$, which is a one-dimensional variable embedded in $d$ dimensions.*

The formal proof is an induction on $d$ and relies on 1) the fact that integer discrete flows embed the discrete random variables into the integer lattice $\mathcal{X} = \mathbb{Z}^d$, which always has a sufficient number of classes (countably infinite), and 2) the observation that two translations are sufficient to map two dimensions into a single dimension with more classes, see Appendix A. For a more intuitive illustration, Figure 1 depicts the type of operations that a single additive bijector can and cannot perform on a three-dimensional probability tensor. In Figure 2 we show that, although sensitive to initialization, an IDF can be trained to factorize the toy example of (5). The first three panels correspond to the data distribution and the intermediate and factorized final distributions produced by the first and second coupling layer respectively. For architecture details see Appendix D.3.

As pointed out by Papamakarios et al. (2019), a discrete flow can only map a uniform base distribution into a uniform data distribution due to its permutation property. Therefore, invertible discrete flows require one-dimensional learnable distributions in order to be able to model all data distributions. Note also that discrete flows as introduced by Tran et al. (2019) are designed for a finite number of classes in $\mathcal{X}$ and therefore *cannot* factorize all distributions. Although not implemented in the original work, this model class can in principle also be extended to embed the data into a space with a larger finite number of possible values than present in the support of the data in order to help alleviate this issue. Since IDFs treat the random variable as integers (with a countably infinite number of classes) it does not have a limited factorization capacity. We therefore explore other directions for potential improvements.

## 5   DOES GRADIENT BIAS HINDER OPTIMIZATION OF IDFS?

In this section we investigate whether gradient bias is a problem when training IDF models. Hoogeboom et al. (2019a) demonstrated that the performance of IDFs can deteriorate when the number of coupling layers is too large, and that this does not occur in the continuous additive counterpart without rounding operators or straight-through gradient estimators. The gradient bias induced by the straight-through estimator was suggested to be the cause.

In our experiments we found that neither stochastic rounding nor replacing the identity function in the straight-through estimator with a soft approximation of the rounding function improved the results. To disentangle the effect of gradient bias of the straight-through estimator and the difficulty of discrete optimization, we have trained IDFs and their continuous additive counterpart (without the rounding operator and without a quantized base distribution) on CIFAR-10. We use different combinations of translation activation functions and backward substitute functions in the straight-through estimator: soft rounding, hard rounding and the identity function. The soft rounding function $\sigma_T$ was modeled with a superposition of scaled sigmoid functions and a temperature $T$ that allows to interpolate between the identity function ($T = 1$) and the rounding function $T = 0$, see Appendix C for more details. The left plot in Figure 3 shows that the continuous model with an identity straight-through

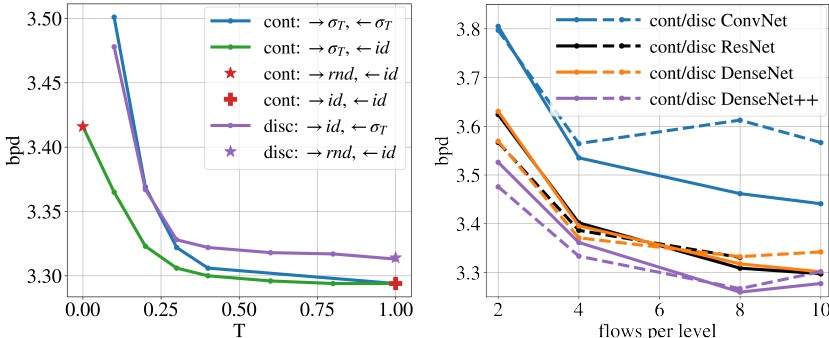

Figure 3: *Left*: Bits per dimension for models trained for 2000 epochs on CIFAR-10. Continuous (cont) and discrete (disc) models are trained with different combinations of translations activation functions ($\rightarrow$) and backward ($\leftarrow$) substitute functions for the straight-through estimators: a soft rounding function ($\sigma_T$), a hard rounding function ($rnd$) and the identity function ($id$). $\sigma_T$ interpolates between the identity function at $T = 0$ and the hard rounding function at $T = 1$. *Right*: Bits per dimension of IDF models trained for 300K iterations ($\pm 960$ epochs) with different architectures for the translation networks in the coupling layers: convolutional neural networks, ResNets and DenseNets. The models have 3 levels. The discrete ResNet model for 10 flows per level was unstable so the result for this model is omitted. The models with the DenseNet architecture correspond to the best models used by Hoogeboom et al. (2019a). The ConvNets and ResNets have approximately equal number of parameters and depth as the DenseNets. Note that the ConvNets used to produce the results in Figure 5 of (Hoogeboom et al., 2019b) are much shallower. The DenseNet++ architectures correspond to the proposed IDF++ model of Section 6.

gradient estimator consistently outperforms the continuous model without gradient bias. The discrete model clearly also does not benefit from a soft-rounding operator in the straight-through estimator, even though this estimator reduces the bias. Note that the continuous model trained with a hard rounding function and identity straight-through estimator ($\rightarrow rnd, \leftarrow id$) performs worse than the discrete model because the continuous model needs to maximize a lower bound to the likelihood.

We numerically study the significance of gradient bias in more detail on an extension of the two-dimensional toy example of Section 4 to 8 bits with a model with two coupling layers. For details on the 8-bit extension of the toy example see Appendix B and for architecture details see Appendix D.3. We study the search directions using finite differences as an approximate gradient vector $\boldsymbol{g}^{\text{fd}}$ with elements $\boldsymbol{g}_i^{\text{fd}} = (L(\theta_i + \epsilon, \boldsymbol{\theta}_{/i}) - L(\theta_i - \epsilon, \boldsymbol{\theta}_{/i}))/2\epsilon$. Here $L$ is the loss function averaged over a single batch of data. For additive continuous flow models, $\boldsymbol{g}^{\text{fd}}$ will approach the true loss gradient $\nabla_{\boldsymbol{\theta}} L$ as $\epsilon \rightarrow 0$. For discrete models, $\boldsymbol{g}^{\text{fd}}$ can be thought of as a linear approximation of the loss landscape in a trust-region of radius $\epsilon$ around the current parameter vector $\boldsymbol{\theta}$.

We compare continuous flow models that are trained using the unbiased gradient $\nabla_{\boldsymbol{\theta}} L$ with discrete flow models that are trained using the straight-through gradient estimator $\boldsymbol{g}^{\text{st}}$. We estimate the agreement between the (approximate) gradient and the finite difference approximation $\boldsymbol{g}^{\text{fd}}$ for varying trust-region size $\epsilon$ at various stages of training and for varying bit-depth of the input data. We compute the agreement of the (approximate) gradient directions with the cosine similarity $\boldsymbol{g} \cdot \boldsymbol{g}^{\text{fd}}/||\boldsymbol{g}||_2||\boldsymbol{g}^{\text{fd}}||_2$, which can be interpreted as the uncentered correlation between the elements of $\boldsymbol{g}$ and $\boldsymbol{g}^{\text{fd}}$. As long as the agreement is consistently positive, performing gradient descent with $\boldsymbol{g}$ is expected to reduce the loss according to the trust-region approximation based on $\boldsymbol{g}^{\text{fd}}$. If the agreement is consistently zero, or even negative, gradient descent with $\boldsymbol{g}$ is not expected to improve the training loss. The agreements are estimated over multiple batches. As the base distribution parameters are not affected by the gradient bias we only consider the gradients for the parameters of the bijectors.

Figure 4 shows the agreement between the finite-difference gradient and the straight-through gradient of the discrete model and the agreement with the real gradient for the continuous model, at initialization and after 3000 training iterations. The straight-through gradient estimator for 8-bit data is always positively correlated with small but finite difference estimates, corroborating our results on CIFAR-10 that the identity straight-through gradient estimator allows for good optimization. For 1-bit data the quality of the straight-through gradient estimator clearly deteriorates and the correlation becomes zero or even negative. This is not surprising as the random variables and translations are in practice modeled on the rescaled grid $\mathbb{Z}/2^{\text{bits}}$, leading to more coarse-grained rounding for lower bits. As lossless compression mostly deals with source data in the higher bit ranges the poor performance

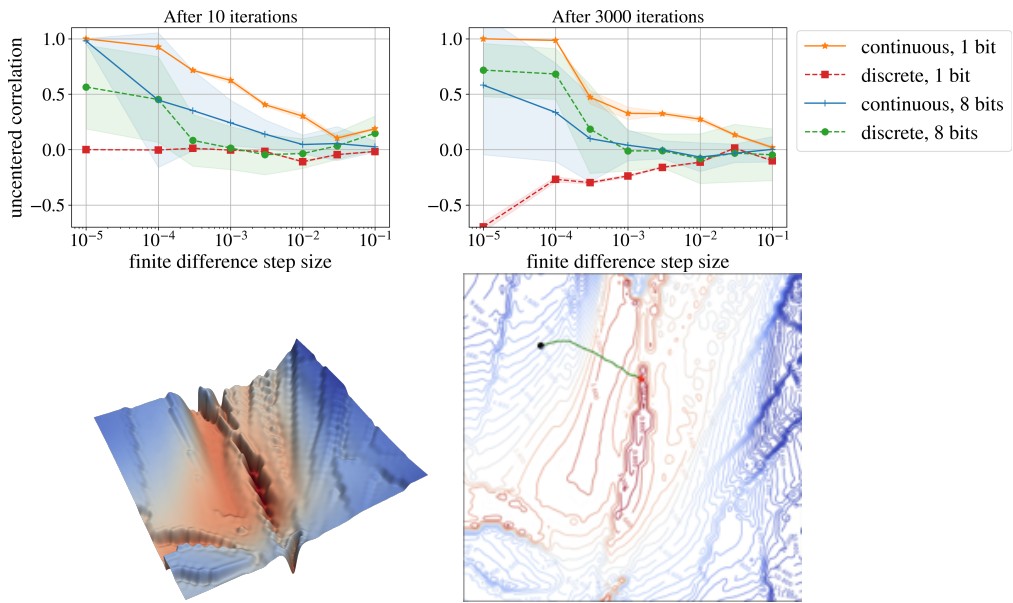

Figure 4: *Top two panels*: Agreement between the approximate (true) gradient and finite difference gradient of the discrete (continuous) model at model initialization (left), and after 3000 training iterations (right). The curves were obtained by averaging over 10 batches. The higher variance for the 8-bit case is due to the larger number of different datapoints, leading to more variance in the data batches. *Bottom two panels*: Visualization of the loss surface (left) and trajectory of the optimizer on the loss surface (right) for the discrete model on the 1-bit toy example, following (Li et al., 2018).

for lower bit data is less relevant. The bottom two panels in Figure 4 show the loss surface and optimization trajectory of a discrete model trained on the 1-bit data, suggesting that the optimization landscape is hard to traverse for low bit-depths. See Appendix B for loss landscapes for 8-bit data and for similar plots for the corresponding continuous models.

Finally, we observed that the performance degradation of IDF as a function of the number of coupling layers is highly dependent on the architecture used for the translation networks in the coupling layers. The right plot in Figure 3 shows the performance of continuous and discrete models with identity straight-through estimators, under different coupling layer architectures: stacked convolutions, ResNets (He et al., 2016) or DenseNets (Huang et al., 2017). All coupling layer architectures are of approximately the same size and depth, and the DenseNet models have the same architecture as the best model of (Hoogeboom et al., 2019b). The performance of the convolutional model clearly deteriorates quicker than the ResNet and DenseNet models.

The results in this section show that gradient bias due to the straight-through estimator is less of a problem than previously suggested. Alternative gradient estimators with less bias do not improve the results, and our analysis on the toy examples shows that for data with large bit depths the straight-through gradient estimates have positive correlation with finite difference estimates. We show that the previously observed performance deterioration as a function of the number of coupling layers is highly dependent on the architecture choice of the coupling layers. This motivates our search for architectural changes that can help improve the performance of IDFs on lossless image compression.

## 6  IMPROVING INTEGER DISCRETE FLOWS FOR LOSSLESS COMPRESSION

In this section we introduce changes to the architecture of IDF that improve its performance. We briefly summarize the IDF architecture, for more detailed information see Appendix D. Similar to other generative flow models, the architecture of IDF contains $L$ levels. Each level $l$ consists of a space-to-depth transformation followed by a sequence of $K$ alternating channel permutations and additive coupling layers. For levels $l = 1, ..., L-1$ the output random variable is split into two pieces, with the second part serving as an input to the next level: $[\boldsymbol{z}^{(l)}, \boldsymbol{y}^{(l)}] = f^{(l,K)} \circ ... \circ f^{(l,1)}(\boldsymbol{y}^{(l-1)})$, with $\boldsymbol{y}^{(0)} = \boldsymbol{x}$. For the last level we simply have $\boldsymbol{z}^{(L)} = f^{(L,K)} \circ ... \circ f^{(L,1)}(\boldsymbol{y}^{(L-1)})$. The combined

$z = [z^{(1)}, ..., z^{(L)}]$ denotes the latent representation of $x$. The distribution of $z$ for 3 levels is then factorized as $p(z) = p(z^{(1)}, z^{(2)}, z^{(3)}) = p(z^{(3)})p(z^{(2)}|y^{(2)})p(z^{(1)}|y^{(1)})$, which is equivalent to $p(z^{(3)})p(z^{(2)}|z^{(3)})p(z^{(1)}|z^{(2)}, z^{(3)})$. The conditional distributions are modeled as discretized logistic distributions and the unconditional $p(z^{(3)})$ is modeled with a mixture of discretized logistics with 5 components. Both the conditioning networks and the pre-quantized translations $t_\theta$ in (4) are modeled with DenseNets (Huang et al., 2017).

The first modification is to invert the channel permutations after every coupling layer. Although inverting the permutation after each translation does not affect the coupling layers of the network, it affects the way the data is split after each level and therefore influences the modeling of the conditional distributions $p(z^{(l)}|y^{(l)})$. By inverting the permutations one ensures that the split into $y^{(l)}$ and $z^{(l)}$ happens along the channel direction of the space-to-depth transformed version of $y^{(l-1)}$; as such $y^{(l)}$ and $z^{(l)}$ retain the spatial correlation structure of the *original* image presumably making it easier to model the conditional distribution $p(z^{(l)}|y^{(l)})$ (see visualization in Figure 10 of Appendix D).

The second change is an adaptation of the rezero trick by Bachlechner et al. (2020). The additive bijectors of (4) are replaced by $[y_a, y_b] = [x_a, x_b + \lfloor \alpha t_\theta(x_a) \rceil]$, where $\alpha$ is a learnable scalar parameter initialized to zero, such that the bijectors are initialized to the identity function. The mean and log-scale of the conditional discretized logistic distributions are parameterized as $\mu = \gamma \nu$, $\log s = \delta \log \sigma$ with $[\nu, \log \sigma] = \text{DenseNet}_\phi(y)$ and $\gamma$ and $\delta$ learnable scalar parameters initialized to zero.

The third modification consists of an alteration in the dense blocks that make up the translation and logistic conditioning DenseNets by introducing group normalization layers (Wu & He, 2018) and switching from ReLU activations (Nair & Hinton, 2010) to Swish activations (Ramachandran et al., 2017):

    IDF:   Conv1x1 → ReLU → Conv3x3 → ReLU

    IDF++:   Conv1x1 → GroupNorm → Swish → Conv3x3 → GroupNorm → Swish

The trainability of deep neural networks such as flow models is strongly dependent on careful initialization and normalization. Identity mapping initialization of flow models, which we achieve by virtue of the rezero trick, is a known technique for improving training stability of these models (Kingma & Dhariwal, 2018). The use of group normalization layers ensures that the dense blocks within the coupling layers receive properly scaled and centered inputs. Combined with the Swish activation function this allows DenseNet activations to remain non-zero and the coupling layers to utilize their capacity more effectively. Empirically, these intuitions are supported by the fact that collectively these modifications allow for the use of a higher base learning rate during training while achieving better results. Finally, during training we also keep track of a slow exponential moving average of the models weights, and use these average weights during evaluation.

Figure 3 shows the performance of models with the proposed modifications (DenseNet++) on the validation set (consisting of 20% of the training set) on CIFAR-10 as a function of flows per level, after 300K iterations. For this new architecture the performance of *both* the continuous and the discrete version of the model deteriorates for 10 flows per level due to overfitting (see Figure 11). These results also show that the modifications allow for a more efficient use of the flow layers: for 300K iterations the performance of the IDF++ model with a DenseNet++ architecture is on par with that of the IDF model of Hoogeboom et al. (2019a) with DenseNet coupling layers.

To further corroborate this point, we have trained an IDF++ model with 4 and 8 flows per level until convergence for CIFAR-10, ImageNet-32 and ImageNet-64, and compared it with the baseline IDF model with 8 flows per level and other related work in Table 1. To make a fair comparison against other methods like local bits-back coding (LBB) by Ho et al. (2019b) we train our final models on the entire training set without holding out part of the training set as a validation set. Although there is no visible difference for the ImageNet-32 and ImageNet-64 datasets, more data does matter for CIFAR-10: it reduces the model's average negative log-likelihood in units of bits per dimension (bpd) on the test set from 3.32 (as reported by Hoogeboom et al. (2019a)) to 3.30 for the baseline IDF model. Table 1 displays the bpd resulting from compressing the latent representations $z$ with a range-based Asymmetric Numerical Systems (rANS) entropy coder (Duda, 2013). Results for hand-designed codecs, models with bits-back coding and models without bits-back coding are shown in separate groups. Where available, the bpd as predicted from the models negative log-likelihood is indicated in brackets.

Table 1: Compression results in bits per dimension (bpd) for IDF++, hand-designed codecs and other deep density estimators based on normalizing flows, super resolution and variational auto-encoders. Where available, the bpd according to the model's negative log-likelihood is indicated in parenthesis. Results with a * are taken from Townsend et al. (2019a), and those with † are taken from Hoogeboom et al. (2019a). The IDF CIFAR-10 result indicated with ** is obtained by our implementation of IDF with 100% of the training data used to train the model. In (Hoogeboom et al., 2019a) only 80% of the training data was used to train the best CIFAR-10 model. All other results are from the original papers.

| Compression models | CIFAR-10 | IMAGENET-32 | IMAGENET-64 |
|---|---|---|---|
| PNG (Boutell & Lane (1997)) | 5.87* | 6.39* | 5.71* |
| JPEG-2000 (Rabbani (2002)) | 5.20† | 6.48† | 5.10† |
| FLIF (Sneyers & Wuille (2016)) | 4.19* | 4.52* | 4.19* |
| BIT-SWAP (Kingma et al. (2019)) | 3.82 (3.78) | 4.50 (4.48) | - |
| HILLOC (Townsend et al. (2019a)) | 3.56 (3.55) | 4.20 (4.18) | 3.90 (3.89) |
| LBB (Ho et al. (2019b)) | 3.12 (3.12) | 3.88 (3.87) | 3.70 (3.70) |
| SREC (Cao et al. (2020)) | - | - | 4.29 |
| IDF (Hoogeboom et al. (2019a)) | 3.32 (3.30)** | 4.18 (4.15) | 3.90 (3.90) |
| IDF++, SMALL: 4 FLOWS PER LEVEL | 3.31 (3.29) | 4.16 (4.14) | 3.85 (3.85) |
| IDF++ | 3.26 (3.24) | 4.12 (4.10) | 3.81 (3.81) |

The results show that, contrary to less-biased gradient estimators, the proposed architecture modifications in IDF++ improve the performance over IDF for 8 flows per level for all datasets. Furthermore, the smaller IDF++ model with only 4 flows per level either performs on par or better than the IDF baseline with 8 flows per level. This effectively reduces the number of parameters and time required for a forward pass through the model by a factor of 2, and demonstrates that the proposed modifications in IDF++ enable more efficient compression without sacrificing performance. For a more detailed ablation on the contributions of the proposed modifications see Appendix E.

## 7    CONCLUSION

In this paper we investigated several potential directions for improvement of integer discrete flows for lossless compression. We analyzed the hypothesis and claims of recent works on the weaknesses of this model class, and we showed that the claim by Papamakarios et al. (2019) that flows for discrete random variables cannot factorize all distributions does not apply to integer discrete flows. Furthermore, we demonstrated that the effect of gradient bias on optimization is less severe than previously thought. Other gradient estimators with less bias do not improve optimization, and a numerical analysis of the direction of the straight-through gradient estimates showed that they correlate well with finite difference estimates for large-bit data. We also showed that the previously observed performance deterioration with increasing depth of the flow model is highly dependent on the architecture of the coupling layers. Motivated by this, we proposed several architecture changes that improve the performance of IDFs on lossless image compression for an equal computational budget. The proposed modifications lead to a more efficient flow-based compression model, as evidenced by our results that show that an IDF++ model with half the number of flows compared to the baseline IDF model performs on par or better.

Although we found that the simple straight-through gradient estimator outperformed all of the other estimators that we considered, future work could consider other estimators inspired by universal quantization and the statistics of quantization noise, similar to the work by Agustsson & Theis (2020). Another important direction for future work is to further reduce the computational complexity of deep density estimators. Although hand-designed codecs such as JPEG-2000 do not compress as well on the datasets we consider, they (de)compress significantly faster and require significantly less memory while not being tuned for each dataset. More work in directions like optimizing for (de)compression speed (Cao et al., 2020) or generalizing learnable compressors to other datasets (Townsend et al., 2019a) is needed to make deep density estimators more practical for source compression.

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

# Supplementary Material of "IDF++: Analyzing and Improving Integer Discrete Flows for Lossless Compression"

## APPENDIX A    FLEXIBILITY OF INTEGER DISCRETE FLOWS

The proof of Lemma 1 can be done in two ways, one is by giving the exact mapping, and the second way is through induction. We include both below as well as the proof for Lemma 2.

*Proof of Lemma 1.* The invertible mapping is given by $y = f(\boldsymbol{x}) = x_1 + K^{(1)}x_2 + K^{(1)}K^{(2)}x_3 + \cdots + \prod_{i=1}^{d-1} K^{(i)}x_d$, with a maximum value of $\left(\prod_{i=1}^{d} K^{(i)}\right) - 1$ and minimum value of zero. Given $y$, each element $x_i$ can be obtained recursively: $x_i = \left(y - \left(\sum_{j=i+1}^{d} K^{j-1}x_j\right)\right)//K^{i-1}$, where $//$ refers to integer division. Through the change of variables formula we know that $p_y(y) = p_{\boldsymbol{x}}(f^{-1}(y))$, which combined with zero padding to embed in $d$ dimensions leads to the desired factorized distribution in $d$ dimensions.    □

*Alternative proof of Lemma 1.* An alternative proof is via induction on $d$.

Base case: Let us start with $d = 2$, we have the random variable $\boldsymbol{x} = (x_1, x_2)^T$ with $x_1 \in \{0, ..., K^{(1)} - 1\}$ and $x_2 \in \{0, ..., K^{(2)} - 1\}$. $\boldsymbol{x}$ is distributed according to $p_{\boldsymbol{x}}(x_1, x_2)$. The following function can map this random variable to a one-dimensional random variable $y$ with $y \in \{0, \ldots, K^{(1)}K^{(2)} - 1\}$: $y = x_1 + K^{(1)}x_2$. Given $y$, $x_1$ and $x_2$ can be recovered through $x_2 = y//K^{(1)}$ and $x_1 = y - K^{(1)}x_2$.

General case: Now let us assume that the lemma holds for $d = n$, and examine the case of $d = n + 1$: $\boldsymbol{x} = (x_1, ..., x_n, x_{n+1})^T$ with $x_i \in \{0, ..., K^{(i)}\}$. Then the following transformation maps to an $n$-dimensional random variable $\boldsymbol{y} = (y_1, \ldots y_n)^T$: $y_i = x_i$ for $i = \{1, \ldots, n-1\}$ and $y_n = x_n + K^{(n)}x_{n+1}$. $\boldsymbol{x}$ can be recovered from $\boldsymbol{y}$ through $x_i = y_i$ for $i = \{1, \ldots, n-1\}$ and $x_{n+1} = y_n//K$, $x_n = y_n - Kx_{n+1}$. We have $y_1, ..., y_{n-1} \in \{0, ..., K^{(i)} - 1\}$ and $y_n \in \{0, ..., K^{(n)}K^{(n-1)} - 1\}$. As we assumed that we can transform an $n$-dimensional random variable to a one-dimensional random variable in an invertible manner, this concludes that we can also do so for the $d = n + 1$ case.    □

*Proof of Lemma 2.* We will prove this via induction on $d$.

Base case: Let us start again with $d = 2$. $\boldsymbol{x} = (x_1, x_2)^T$ with $x_i \in \{0, \ldots, K^{(i)}\}$ can be transformed into a random variable $\boldsymbol{y} = (y_1, 0)^T$ in an invertible manner with two translation operations of the form $z_a = x_a$, $z_b = x_b + \lfloor t(x_a) \rceil$, where either $(a, b) = (1, 2)$ or $(a, b) = (2, 1)$. The first translation is given by $z_1 = x_1 + K^{(1)}x_2$, $z_2 = x_2$, followed by $y_1 = z_1 = x_1 + K^{(1)}x_2$, $y_2 = x_2 - (x_1 + K^{(1)}x_2)//K^{(1)} = x_2 - x_2 = 0$.

General case: Let us assume translations are sufficient for $d = n$. We can then transform a $d = n + 1$ dimensional variable $\boldsymbol{x} = (x_1, \ldots, x_n, x_{n+1})^T$ in an invertible manner to a random variable that is effectively $n$-dimensional: $\boldsymbol{y} = (y_1, \ldots, y_n, 0)$ with two translations operations: The first translation is:

$$z_i = x_i \text{ for } i \in \{1, \ldots, n-1, n+1\}$$
$$z_n = x_n + K^{(n)}x_{n+1}, \tag{7}$$

followed by the second translation:

$$y_i = z_i \text{ for } i \in \{1, \ldots, n\}$$
$$y_{n+1} = z_{n+1} - z_n//K^{(n)} = x_{n+1} - (x_n + K^{(n)}x_{n+1})//K^{(n)} = x_{n+1} - x_{n+1} = 0. \tag{8}$$

Which results in $y_i = x_i$ for $i \in \{1, ..., n-1\}$ and $y_n = x_n + K^{(n)}x_{n+1}$. As we assumed integer translations were sufficient to map an $n$-dimensional variable like $(y_1, \ldots, y_n)^T$ to a one-dimensional random variable embedded in $n$ dimensions in an invertible manner, we have now proven that this also holds for $d = n + 1$.    □

## APPENDIX B    LEARNING TOY EXAMPLES

### B.1    GRADIENT BIAS ON THE TOY EXAMPLE

The toy example that is discussed in Section 4 has a probability distribution with nonzero probabilities for $x_i \in \{0, 1\}$. With only two values per dimension with nonzero probability the data is effectively 1-bit quantized. Here we use an extension of this toy example for input data that is quantized to a higher bit-depth. This can be done by modeling the probability masses with log-linearly spaced logits in the interval $(0, 1)$. The resulting distributions for several bit-depths are shown in Figure 5.

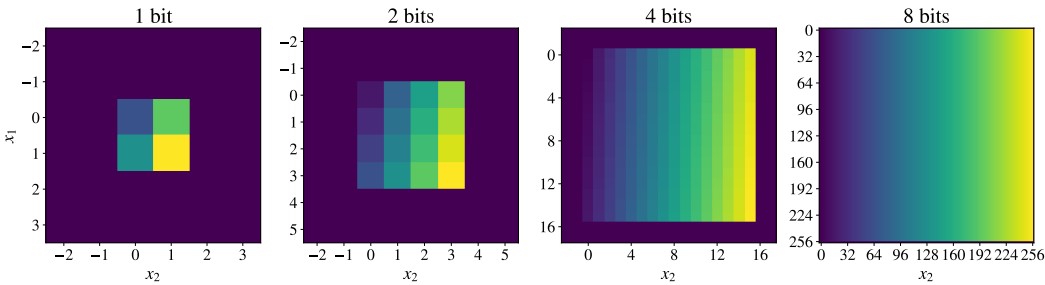

Figure 5: probability distribution of the extension of the toy example of Eq. (5) to more bits.

### B.2    LOSS LANDSCAPE AND OPTIMIZER TRAJECTORY

In Section 5, we presented the visualization of the loss landscape and the optimization trajectory of the *discrete* model on a toy example with 1-bit data. The details on how these plots were obtained will be explained in this section. Navigating through a loss landscape that is affected by quantization operators can be challenging due to discontinuities. To illustrate this effect, we visualize the loss landscape and the optimization path for the discrete and continuous models trained on the 1-bit and 8-bit toy examples. We use the method proposed by Li et al. (2018), where for given model parameters $\boldsymbol{\theta}^*$, we choose two direction vectors $\boldsymbol{\theta}_1$ and $\boldsymbol{\theta}_2$ and plot the value of $f(\alpha, \beta) = \mathcal{L}(\boldsymbol{\theta}^* + \alpha\boldsymbol{\theta}_1 + \beta\boldsymbol{\theta}_2)$. To choose the direction vectors, we use model parameters at different stages of training. Let $\boldsymbol{\theta}_i^*$ be model parameters after iteration $i$. Given the set of parameters for $n$ iterations, we apply PCA to the matrix $\boldsymbol{M} = [\boldsymbol{\theta}_0^* - \boldsymbol{\theta}_n^*, \ldots, \boldsymbol{\theta}_{n-1}^* - \boldsymbol{\theta}_n^*]$ and select the two most explanatory directions as $\boldsymbol{\theta}_1$ and $\boldsymbol{\theta}_2$.

The results for the 1-bit toy example are shown in the left two panels of Figure 4 for the discrete flow model. Figure 6 shows similar plots for the continuous case, demonstrating a much smoother loss landscape. Figure 4 suggests that discrete models with unlucky initializations can easily end up in a local minimum from which it is hard to escape due to sharp cliffs in the loss landscape. Figures 7 and 8 show a significantly less pronounced difference between the loss landscape of the continuous and discrete model for the 8-bit toy example. This supports our observation in Section 5 that the gradient bias is less of an issue at higher bit-depths.

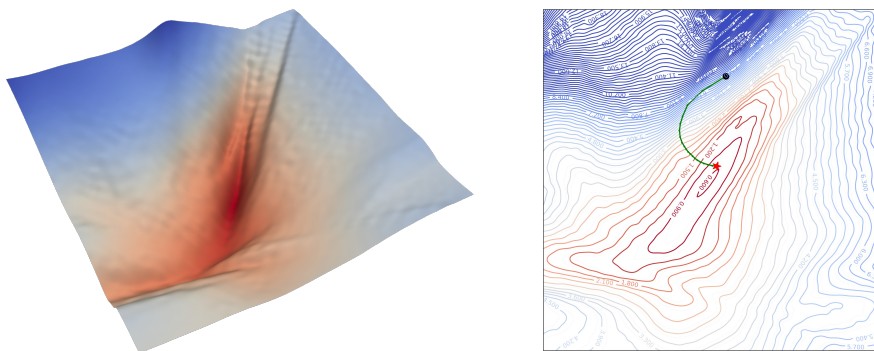

Figure 6: Visualization of the trajectory of the optimization on the loss surface (Li et al., 2018) for the continuous model for the toy example with 1-bit data.

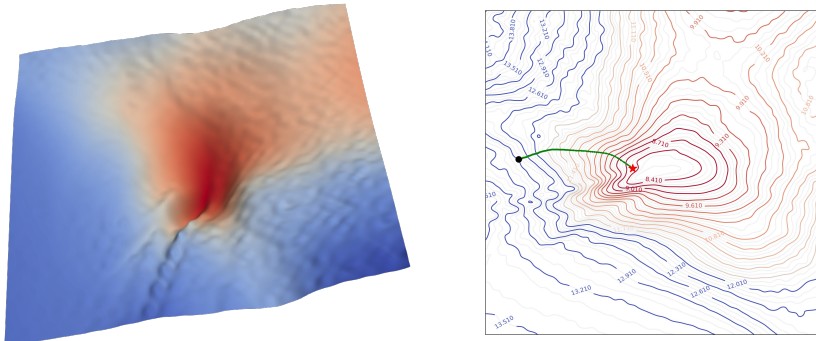

Figure 7: Visualization of the trajectory of the optimization on the loss surface (Li et al., 2018) for the discrete model for the toy example with 8-bit data.

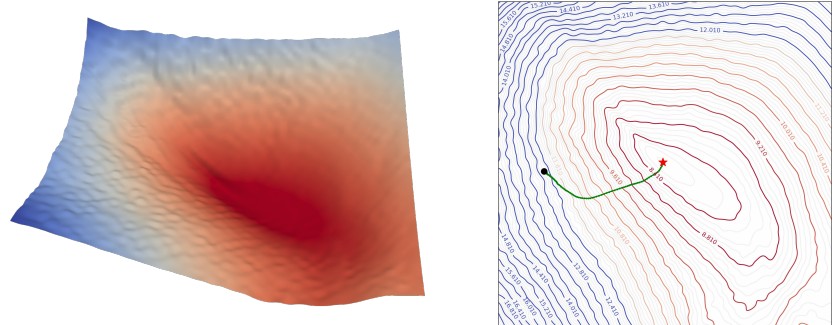

Figure 8: Visualization of the trajectory of the optimization on the loss surface (Li et al., 2018) for the continuous model for the toy example with 8-bit data.

## APPENDIX C    SOFT ROUNDING FUNCTION

The soft rounding function $\sigma_T$ that is used in Section 5 is given by

$$\sigma_T(x) = \lfloor x \rfloor + \frac{\sigma\left(\frac{2}{T}(x - \lfloor x \rfloor) - \frac{1}{T}\right)}{\sigma\left(\frac{1}{T}\right) - \sigma\left(-\frac{1}{T}\right)} \ . \tag{9}$$

The soft rounding function is depicted in Figure 9 for different temperatures.

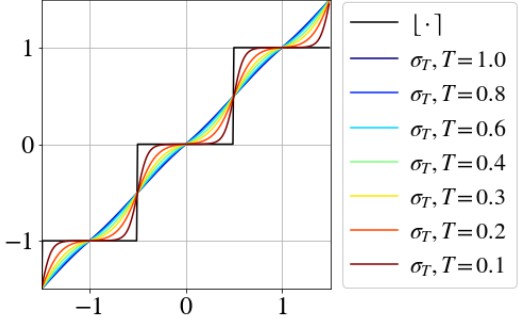

Figure 9: Soft rounding function $\sigma_T$ for different temperatures $T$. In the limit $T = 0$ the soft rounding function reduces to the hard rounding function $\lfloor \cdot \rceil$. At $T = 1$ the soft rounding function is indistinguishable from the identity function.

## APPENDIX D  ARCHITECTURES

### D.1  ARCHITECTURE AND TRAINING DETAILS OF IDF

The architecture of IDF consists of $L$ levels, each consisting of an alternating sequence of channel permutations and additive bijectors. More specifically, the architecture of level $l$ of IDF is built up as follows:

$$\boldsymbol{y}^{(l-1)} \to \text{space-to-depth} \to \overbrace{\{\text{Permute} \to \text{Additive transform}\}}^{\times K} \to [\boldsymbol{z}^{(l)}, \boldsymbol{y}^{(l)}].$$

Except for the last level, the output random variable is split into two equal halves $[\boldsymbol{z}^{(l)}, \boldsymbol{y}^{(l)}]$, where only $\boldsymbol{y}^{(l)}$ is transformed by the next level. Furthermore, $\boldsymbol{y}^{(0)} = \boldsymbol{x}$ and $\boldsymbol{z} = [\boldsymbol{z}^{(1)}, \ldots, \boldsymbol{z}^{(L)}]$ denotes the latent representation of $\boldsymbol{x}$. Inside the additive transformations of (4) the prequantized translations are modeled using DenseNets (Huang et al., 2017) with dense blocks of the following structure:

$$\text{Dense block:}\quad \text{Conv1x1} \to \text{ReLU} \to \text{Conv3x3} \to \text{ReLU}.$$

All DenseNets have a depth of 12 blocks and 512 channels. The additive bijector splits the random variable into two parts along the channel dimension with splitting fractions $3/4$ and $1/4$ for the untransformed and transformed parts of the random variable. After translating, the resulting variables are concatenated again along the channel axis.

The distribution of $\boldsymbol{z}$ for $L$ levels is factorized as $p(\boldsymbol{z}) = p(\boldsymbol{z}^{(L)})p(\boldsymbol{z}^{(L-1)}|\boldsymbol{y}^{(L-1)}) \ldots p(\boldsymbol{z}^{(1)}|\boldsymbol{y}^{(1)})$, which is equivalent to $p(\boldsymbol{z}^{(L)})p(\boldsymbol{z}^{(L-1)}|\boldsymbol{z}^{(L)}) \ldots p(\boldsymbol{z}^{(1)}|\boldsymbol{z}^{(2)}, \ldots \boldsymbol{z}^{(L)})$. All conditional distributions are modeled with discretized logistic distributions. The unconditional distribution $p(\boldsymbol{z}^{(L)})$ is modeled with a mixture of discretized logistics with five components. The log-scale and mean of the conditional logistic distributions are modeled as the outputs of DenseNets with the same structure as the DenseNets of the prequantized translations: $[\boldsymbol{\nu}, \log \boldsymbol{\sigma}] = \text{DenseNet}_\phi(\boldsymbol{y})$. Note that instead of modeling the random variables as integers ($\boldsymbol{x} \in \mathbb{Z}^d$), they are modeled as discrete random variables on a grid with bin-width $1/256$ ($\boldsymbol{x} \in \mathbb{Z}^d/256$).

The model is trained with the Adamax optimizer (Kingma & Ba, 2014) with an exponential learning rate schedule with base learning rate equal to $1 \times 10^{-3}$ and a linear warmup phase of 10 epochs. See Table 2 for more details on the learning rate decay, the number of levels, the batch size and the number of epochs used for training.

Table 2: Architecture and training settings for IDF. Table adapted from Hoogeboom et al. (2019a). Note that we used a batch size of 128 for CIFAR-10 instead of 256 as used in the original work, while still reproducing the same number as reported by Hoogeboom et al. (2019a). For ImageNet-32 and ImageNet-64 the batch sizes are the same as in the original work (Hoogeboom et al., 2019a).

| Dataset | Levels $L$ | Batch size | Learning rate decay | Epochs |
|---|---|---|---|---|
| CIFAR-10 | 3 | 128 | 0.999 | 2000 |
| ImageNet-32 | 3 | 256 | 0.99 | 100 |
| ImageNet-64 | 4 | 64 | 0.99 | 10 |

Range-based Asymmetric Numerical Systems (rANS) (Duda, 2009; 2013; Townsend, 2020) is used for lossless compression of the latent variables $\boldsymbol{z} = [\boldsymbol{z}^{(1)}, \ldots, \boldsymbol{z}^{(L)}]$ by using the probability distribution $p(\boldsymbol{z}) = p(\boldsymbol{z}^{(L)})p(\boldsymbol{z}^{(L-1)}|\boldsymbol{y}^{(L-1)}) \ldots p(\boldsymbol{z}^{(1)}|\boldsymbol{y}^{(1)})$ corresponding to the model's multi-level structure.

### D.2  ARCHITECTURE AND TRAINING DETAILS OF IDF++

In IDF++, each of the $K$ blocks containing a permutation and additive bijector has an additional inverse channel permutation to ensure that the output random variable has the same spatial and channel ordering as the input random variable:

$$\boldsymbol{y}^{(l-1)} \to \text{space-to-depth} \to \overbrace{\{\text{Permute} \to \text{Additive transform} \to \text{Inverse permute}\}}^{\times K} \to [\boldsymbol{z}^{(l)}, \boldsymbol{y}^{(l)}].$$

The inversion of each permutation inside a level ensures that the splitting at the end of each level is done along the channel dimension of a space-to-depth transformed image, enabling the conditional distributions to be able to use the spatial correlation between pixels. See Figure 10 for a visualization. The dense blocks of the DenseNets for the prequantized translations and the conditional discretized logistics have additional group normalization layers (Wu & He, 2018) and Swish activations (Ramachandran et al., 2017) instead of ReLU activations (Nair & Hinton, 2010):

Dense block:   Conv1x1 $\rightarrow$ GroupNorm $\rightarrow$ Swish $\rightarrow$ Conv3x3 $\rightarrow$ GroupNorm $\rightarrow$ Swish.

The number of groups for each group normalization layer are determined as follows: if the number of channels of the group normalization layer is divisible by 3 then 3 groups are used, else if it is divisible by 2 then 2 groups are used, and finally if it is neither divisible by 3 or 2 then a single group is used.

The additive bijectors in (4) are adjusted to include a learnable scalar parameter that ensures initialization to the identity operator, similar to the rezero trick by Bachlechner et al. (2020):

$$\begin{bmatrix} \boldsymbol{y}_a \\ \boldsymbol{y}_b \end{bmatrix} = \begin{bmatrix} \boldsymbol{x}_a \\ \boldsymbol{x}_b + \lfloor \boldsymbol{t}_{\boldsymbol{\theta}}(\alpha \boldsymbol{x}_a) \rceil \end{bmatrix} . \tag{10}$$

Here $\alpha$ is a learnable scalar parameter initialized to zero. The mean and log-scale of the conditional discretized logistic distributions are parameterized as $\boldsymbol{\mu} = \gamma \boldsymbol{\nu}$, $\log \boldsymbol{s} = \delta \log \boldsymbol{\sigma}$ with $[\boldsymbol{\nu}, \log \boldsymbol{\sigma}] = \mathrm{DenseNet}_{\theta}(\boldsymbol{y})$. $\gamma$ and $\delta$ are learnable scalar parameters initialized to zero, such that the scale is initialized to one and the mean is initialized to zero.

The combination of the rezero trick and the group normalization layers allows us to use a larger base learning rate of $2 \times 10^{-3}$ in the exponential decayed learning rate schedule. For CIFAR-10, the IDF++

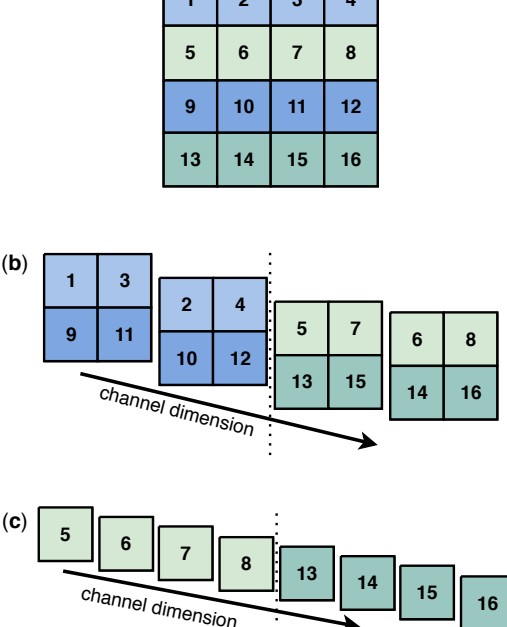

Figure 10: Schematic visualization of the effect of inverse permutations on the IDF++ architecture. In a multi-level IDF++ model the original image (**a**) undergoes a space-to-depth transformation (**b**), after which at the end of the first level half of the dimensions (blue tones) are factored out and their conditional distribution based on the remaining dimension (green tones) is learned. This procedure is repeated for the remaining dimensions (**c**) in the next level where again half the dimensions (light green) are factored out and their conditional distribution is modelled based on the remaining dimensions (dark green). By inverting permutations introduced in the coupling layers of each level we guarantee that the spatial structure of the original image is preserved at all levels. Specifically, with this structure, the factored out and remaining dimensions correspond to nearby rows of the original image. We expect that the spatial correlation present in nearby rows is effectively leveraged by the conditional distributions parameterized with convolutions, leading to better results.

model with 8 flows per level was trained for 1400 epochs instead of 2000 epochs as overfitting was observed for more epochs. We evaluated all variations of IDF with an exponential moving average of the trainable parameters with a decay rate of 0.9999.

### D.3 ARCHITECTURE FOR TOY EXAMPLE MODELS

The models for the toy examples in Section 4 and 5 have a single level with an unconditional prior given by a mixture of logistics with five components. The models have two coupling layers with DenseNets as translation neural networks. The dense blocks have the following structure

$$\text{Dense block:} \quad \text{Conv1x1} \rightarrow \text{LayerNorm} \rightarrow \text{ReLU} \rightarrow \text{Conv1x1} \rightarrow \text{LayerNorm} \rightarrow \text{ReLU}.$$

We found the inclusion of LayerNorm (Ba et al., 2016) to help make the model less sensitive to initialization. Because the DenseNets in these models are fairly shallow we did not see a performance difference between LayerNorm and GroupNorm. The DenseNets have a depth of 4 blocks and 32 channels. The datapoints are treated as images of shape $h \times w \times c = 1 \times 1 \times 2$. The additive bijectors split the random variable into two parts along the channel dimension. After translating, the resulting variables are concatenated again along the channel axis. In between the coupling layers the channels are reversed to allow for alternating conditioning. The learning rates for the prior parameters and coupling layers are set to $1 \times 10^{-3}$ and $1 \times 10^{-4}$ respectively and we used a batchsize of 128.

### D.4 DATASETS

The training set of CIFAR-10 consists of 50000 images and the test set contains 10000 images. ImageNet-32 and ImageNet-64 contain approximately 1250000 train images and 50000 test images. While we used a validation set that was held out from the respective training sets for model development, we trained our final models on the entire training set for each dataset. The effect of this is discussed in Section E. We follow Hoogeboom et al. (2019a) and augment the CIFAR-10 dataset with horizontal flipping, reflect-padding and random cropping during training. No augmentation is used for ImageNet-32 and ImageNet-64.

### D.5 HARDWARE AND SOFTWARE

We implemented our models in TensorFlow (Abadi et al., 2015). All experiments were run with 8 NVIDIA V100 GPUs.

## APPENDIX E ABLATION STUDIES

Table 3 shows the contributions of the modifications proposed in Section 6 for IDF++. As Hoogeboom et al. (2019a) only use 80% of the training data for CIFAR-10, we also show the influence of using the entire train set for training, as opposed to keeping 20% held out as a validation set. Note that the result for the original IDF model on 80% of the training data reproduces the result of Hoogeboom et al. (2019a). The held out validation set for ImageNet-32 and ImageNet-64 was much smaller and we noticed no significant change in performance when training on the entire training set for these

Table 3: Ablation study of the modifications of IDF for the CIFAR-10 dataset. The second column indicates the percentage of the dataset's original train set that was used to train the model. EMA = exponential moving average. The reported numbers are theoretically achievable bits per dimension.

| IDF variations | % Train data | CIFAR-10 |
|---|---|---|
| IDF | 80% | 3.322 |
| IDF | 100% | 3.298 |
| IDF + EMA | 100% | 3.291 |
| IDF + EMA + rezero + lr $\times$ 2 | 100% | 3.262 |
| IDF + EMA + rezero + lr $\times$ 2 + invert perm | 100% | 3.255 |
| IDF + EMA + rezero + lr $\times$ 2 + invert perm + groupnorm (**IDF++**) | 100% | 3.241 |

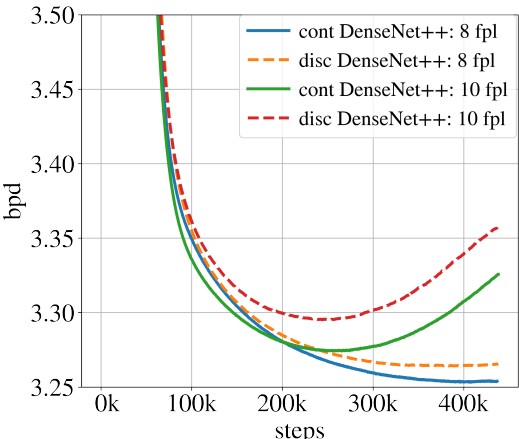

Figure 11: The performance of IDF++ models and their continuous counterparts on the validation set for 8 and 10 flows per level (fpl). The models are trained on 80% of the training data.

datasets. All models without group normalization were trained for 2000 epochs. The final IDF++ model, which includes group normalization layers, was trained for 1400 epochs to avoid overfitting.

In Figure 11 the validation bpd is shown for IDF++ models and their continuous counterparts for 8 and 10 flows per level. The plot shows that both the continuous and the discrete model with 10 flows per level start to overfit.

