# OpenReview forum: "IDF++: Analyzing and Improving Integer Discrete Flows for Lossless Compression"
_ICLR.cc/2021/Conference — ICLR 2021 Poster_

### Official Review · AnonReviewer1 · 2020-10-25

**Rating:** 7
**Confidence:** 4

**Review:**

This paper investigated the weakness of Integer Discrete Flow (IDF) and improved it for the lossless compression task.
 Specifically, the contributions of this paper are:

1. The authors theoretically analyzed the flexibility of normalizing flows for discrete random variables, showing that normalizing flows for discrete random are capable of mapping from a distribution with dependencies across all dimensions to a distribution which is fully factorized.  By relaxing the assumption in Papamakarios et al. (2019) that the domain of the invertible function is the same as the original random variables, the authors denied the claim in Papamakarios et al. (2019) invertible flows for discrete random variables cannot map all distributions to a factorized base distribution (Lemma 1). The authors further demonstrated that the additive coupling layers in IDF is sufficient to model such mapping (Lemma 2).

2. The authors empirically discussed the potential influence of gradient bias on training IDFs, showing that the straight-through estimator is not the cause of gradient bias. Furthermore, the  influence of gradient bias is highly dependent on architecture choices.

3. Based on the previous analysis, the authors proposed an improved architecture for IDF, named IDF++. Experiments on three benchmarks -- CIFAR-10, ImageNet-32 and ImageNet-64 -- show that IDF++ outperforms IDF on all the three benchmarks.

Overall, this paper is well-written and well-motivated. Experimental details are provided to reproduce the results. Ablation studies are conducted to help analyze the effect of different modifications of IDF++.

---

> ### Author Response · Authors · 2020-11-17
> **Response to review by AnonReviewer1**
>
> We thank the reviewer for the positive feedback. Although the reviewer’s comments did not prompt for additional experimental results, we would like to highlight the following additional experiments that we have performed, which we believe further strengthens our contributions:
>
> * We studied the performance of an IDF++ model as a function of flows per level. The results show that IDF++ (with the proposed alterations of the DenseNet architecture, indicated with DenseNet++) can learn better models with a restricted depth as compared to models with other coupling layer architectures. See the updated Figure 3.
> * We show that a converged IDF++ model with 4 flows per level can perform on par with or better than an IDF model with 8 flows per level on CIFAR-10, ImageNet-32 and ImageNet-64. This is an important step in the direction of making flow-based models more efficient data compressors, since this effectively reduces the number of parameters by a factor of 2 and also reduces the run time of the model. See the updated results in Table 1.
>
> Further comments on these results are welcome.

---

> > ### Comment · AnonReviewer1 · 2020-11-18
> > **Feedback to response**
> >
> > Dear Authors, Thank you very much for providing detailed responses. I have also read your responses to other reviewers.
> > The additional experiments make the comparison more comprehensive and clearer, which are highly appreciated.
> >
> > I decide to keep my initial score and would like to see the submission to be accepted.

---

### Official Review · AnonReviewer4 · 2020-10-27
**Some nice theoretical and practical discussion of normalizing flows for discrete data.**

**Rating:** 7
**Confidence:** 4

**Review:**

#### DESCRIPTION
This paper (i) notes that discrete flows are not limited in representation ability when the discrete data is embedded in the integer lattice, (ii) demonstrates that bias due to the straight-through gradient estimator is not as severe as previously thought, and (iii) investigates some architectural tweaks for integer discrete flows which lead to better performance.

#### DISCUSSION
I suppose the degree to which the claim in Papamakarios et al 2019 concerning the representational ability of discrete flows can be considered 'not correct' largely depends on whether you decide to initially embed your discrete data in the integer lattice (Z^D) before formulating a model. If you view a discrete flow on e.g. 8-bit images as a bijection on the support of this data (Z_{2^8}^{D} in the case of 8-bit images), then the claim of Papamakarios et al is correct. I'd argue this is the implicit assumption, since the language is centred around a view of discrete flows as permutations, which are by definition bijections. On the other hand, if you begin by embedding discrete data in the integer lattice before formulating a model as a bijection on this integer lattice, then the same limit on representational ability doesn't hold, as you show in the example and in Lemma 1. My point here is that the distinction made by this paper is useful, but not mutually exclusive with the discussion in Papamakarios et al.

The analysis of gradient bias is good, and works to dispel the result I had taken from Hoogeboom et al 2019 regarding the limitations of deep IDFs. It's maybe not very comforting that performance degradation seems to rely on careful choice of architecture for the nets parameterizing the translations though, and it's not immediately clear to me why a simple CNN deteriorates more quickly -- do you have any explanation for why this might be the case?

Even though it's maybe expected, it's also nice to see that careful consideration of initialization in the nets parameterizing the translations also leads to better performance.

Finally, the experimental result on image data are fine, but I'm not necessarily convinced discrete flows will see practical use a compression tool just yet, as you mention in your conclusion.

#### EXTRA NOTES
Figure 1 is neat and is quite a good visual aid. Despite lacking the same production quality, I also found the 2D grid in eq. 6 helpful for conveying the concept, and a generalization of this 2D grid might have provided a nice geometric proof of Lemma 1.

Because the discretized logistic base distributions are used to parameterize a distribution over the integers rather than the usual 8-bit range, do you find there are any numerical issues with large outputs from the model, or accurately evaluating the CDF for these possible outliers?

Figures 3 & 4: Could both be made larger, with increased label, tick, and legend font sizes. Also (nitpick), "The right two panels in Figure 4 show the loss surface and optimization trajectory of a discrete model trained on the 1-bit data, showing that the optimization landscape is hard to traverse for low bit-depths" -- do those panels really show it definitively, or just suggest it?

#### CONCLUSION
Overall I think the contributions are worthwhile to have in the literature, and would like to see the paper accepted.

---

> ### Author Response · Authors · 2020-11-17
> **Response to review by AnonReviewer4**
>
> We thank the reviewer for their time and constructive feedback. We appreciate that the reviewer mentions the gradient bias analysis, the flexibility analysis and the architecture changes as positive contributions of the paper. Below we respond more in depth to the reviewer’s comments and we indicate which comments are translated into changes in our submission.
>
> **Discussion about flexibility and claims by Papamakarios et al.**
> We agree that the discussion on the flexibility of discrete flows hinges on the choice of space in which the data is embedded. For completeness, we would like to emphasize that it is not necessary for this space to be the integer lattice. A space with an exponentially large (in data dimension) but finite number of possible values is also sufficient. We appreciate how you phrased the distinction between our results and those of Papamakarios et al. and we have used this to re-word our statement on “refuting the claim” in all relevant sections. We are now stating that the limitations for discrete flows can be overcome by embedding the discrete data into a larger set of possible values, and that this is naturally the case for integer discrete flows.
>
> **Gradient bias analysis and the performance of CNN coupling layers**
> We are unsure why exactly the model with CNN-based coupling layers deteriorates more quickly than models with either ResNets or DenseNets. Perhaps the combination of deep flow models and the skip connections inside the ResNet and DenseNet coupling layers makes learning easier, but since we lack more evidence to support this we don’t want to speculate too much.
>
> **Additional experimental results**
> We have included the following new results:
> * We compute the performance of an IDF++ model as a function of flows per level, similar to our analysis of an IDF model with a ConvNet, ResNet and DenseNet architecture. This shows that IDF++ (with the proposed changes in the DenseNet architecture, indicated with DenseNet++) can learn better models with a restricted depth. See the updated Figure 3.
> * By training an IDF++ model with 4 flows per level to convergence, we demonstrate that this smaller model performs on par with or better than an IDF model with 8 flows per level for CIFAR-10, ImageNet-32 and ImageNet64. This effectively reduces the number of parameters by a factor of 2 and also reduces the run time of the model. See the updated Table 1.
>
> We believe that this reduction in memory and runtime is a useful contribution towards making neural network compressors based on flow models more practical.
>
> **Extra notes: numerical stability of discretized logistic for large values**
> Following Hoogeboom et al. we model the integer values on the $\mathbb Z^d/2^8$ grid to avoid numerical issues. We have not encountered any numerical issues with evaluating the CDF of the logistic distribution when rescaling the data to this grid.
>
> **Extra notes: Figures 3 & 4**
> We have taken the reviewer’s suggestions into account and increased the font sizes. We have also changed the caption of Figure 4 to “suggesting”.

---

> > ### Comment · AnonReviewer4 · 2020-11-18
> > **Thanks**
> >
> > Thanks for the response.
> >
> > I think the point about flexibility and the choice of embedding is enough for me to want this paper in the literature -- it's a point I hadn't considered before, and all the more worthwhile considering the explicit discussion of limitations of discrete flows in Papamakarios et al 2019.
> >
> > The remaining contributions are also useful in their own right.
> >
> > I'm happy for the submission to be accepted.

---

### Official Review · AnonReviewer2 · 2020-10-28
**While the paper contains interesting observations, the proposed approach seems incremental and lacks clear insight that can be generalized.**

**Rating:** 6
**Confidence:** 4

**Review:**

**Update after author response**
I thank the reviewers for their response.  I appreciated the more careful discussion of the discrete claims (as other reviewers also noted). I also appreciated the efforts to give more justification for your architecture changes.  While the actual experimental results still seem incremental in terms of raw performance, I think the other contributions of the paper are solid and worth having in the literature.  Thus, I've updated my score.

**Summary**
This paper focuses on lossless compression using integer discrete flows (IDF).  The paper explores the theoretical flexibility of IDF models showing that (with a countably infinite number of possible values for one dimension) any distribution can be factorized.  The paper also explores the bias in the gradient estimates and gives evidence that architecture seems to be more important than unbiased gradient estimators.  The paper proposes a few architecture changes including inverting channels, initializing transform to identity, initializing base distribution to standard logistic distribution, and using GroupNorm and Swish layers.  Finally, the paper shows better compression results than the original IDF model across CIFAR-10, ImageNet-32 and ImageNet-64.

**Strengths:**
- Investigates the potential gradient bias issue of IDFs and demonstrates that this may not be the core issue. (a very nice observation)
- Suggests architecture changes and shows improved performance compared to baseline IDF.
- Nice illustrations of discrete transformations.

**Weaknesses:**
- Claims about "refuting" the claim in [Papamakarios et al. 2018] seem too strong. It seems that Papamakarios et al. [2018] had a restricted case in mind of finite discrete values. And the point still seems interesting for finite discrete values (e.g., binary data).  The first theoretical result in this paper (Lemma 1) seems to be a still interesting but an almost obvious result that if you allow for infinite number of discrete values you can factorize any distribution (i.e., by just putting all possible non-zero configurations along one dimension). Again, while it is interesting and useful to discuss this, a more nuanced discussion is likely needed.  And this should be seen not as refuting the claim but as showing that relaxing one of the assumptions (i.e., finite to countably infinite values) allows for the potential of an arbitrarily flexible distribution.  There should also be a discussion that this transformation could inherently be bad for estimation since it would require the estimation of an exponential number of values.  Thus, the theoretical result may not be practically useful.

- Architecture changes seem to lack clear motivation and insight. While I appreciated the motivation that the limitation may not be gradient bias, I did not understand the reasons for the architecture changes.  Insights into why these architecture changes lead to an improvement in performance are critical.  Why is Swish better than ReLU in this context?  Why is GroupNorm really needed?  Without deeper insights, it is challenging to build off of this work or generalize this work to other contexts.

- The results seem incremental.  The results, while better than IDF, seem relatively incremental (-0.08,-0.06,-0.09).  I appreciated the context of other compression models but it seemed that IDF++ only marginally improves over the baseline IDF.

**Other comments or questions**
- The gradient-bias experimental section is a bit hard to parse.  A rewrite for clarity and simplicity would help the paper.

- Wouldn't finite difference be very expensive to compute?  This would require $O(|\theta|)$ number of model evaluations compared to $O(1)$ model evaluations for gradient estimates. Was this experiment only on a very small dataset and model?

- Why does inverting the channels help?  This still seems unclear to me since it seems to be a trivial modification though I'm probably missing something.

- What does "We furthermore use an exponential moving average for evaluation" mean? Is this related to GroupNorm or  to calculating the log likelihood/BPD?

---

> ### Author Response · Authors · 2020-11-17
> **Response 1/2 to review by AnonReviewer2**
>
> We thank the reviewer for their time and constructive comments. We appreciate that they see value in our investigation of the potential gradient bias issues and the illustrations of discrete transformations. Below we first respond to the main weaknesses raised by the reviewer, followed by responses to the other comments and questions. We hope the additional experiments and mentioned clarifications in the paper can alleviate the main concerns of the reviewer.
>
> **Main weakness 1: discussion about flexibility and claims by Papamakarios et al.**
>
> For completeness, we would first like to comment on a few points made here by the reviewer:
> * Section 5.3 of Papamakarios et al. does not state that they have the restricted case in mind of finite discrete values. They first describe what a bijective discrete flow is for a general discrete set, and then, as concrete examples, give integer discrete flows (indicating also that the discrete set here thus corresponds to integers, which is not a finite set) and discrete flows (by Tran et al. with a finite set of values).
> * Lemma 1 does not state we need an infinite number of values. It states that an exponential number is sufficient (but not necessary). An infinite number is then also sufficient. We are thus not relaxing the assumption of a finite number of values to an infinite one. Our goal is to convey that if you embed data with a finite number of possible values into a space with a sufficiently large (yet finite) number of values, you can factorize any distribution. The case of a countably infinite space (like integers) is then naturally also sufficient.
> * Particularly, we explain that the binary example that Papamakarios et al. consider (the toy example in Section 4), can be solved by embedding the binary data in space with 4 possible discrete values for each dimension.
>
> We appreciate the comments made by the reviewer on Section 4 and hope that the following changes can alleviate the reviewer’s concern on this aspect of the paper:
>
> * We have re-worded our statements in the abstract, introduction, Section 4 and the conclusion on refuting the claim by Papamakarios et al. by using the phrasing suggested by AnonReviewer 4. We are now stating that the limitations for discrete flows as mentioned by Papamakarios et al. can be overcome by embedding the discrete data into a larger set of possible values, and that this is naturally the case for integer discrete flows.
>
> **Main weakness 2: insights into architecture and motivation**
> The main motivation for the changes in the architecture is the fact that optimizing deep flow models can be difficult and is sensitive to initialization, as also mentioned by AnonReviewer4. For instance, the rezero trick ensures that the coupling layers are initialized to the identity functions, and the group normalization layers ensure that the layers inside the DenseNet coupling networks receive properly scaled and centered inputs. The potential reason why inverting the permutations could be beneficial is explained below in the response to “other comments and questions”.
> We agree that this motivation is not described in the submitted version of this paper, and we have included it in the updated version.
>
> **Main weakness 3: incremental results**
> As per the suggestion of AnonReviewer2 we have performed additional experiments. We have computed the performance of IDF++ for several numbers of flows per level, and these new results show that an IDF++ model with 4 flows per level can perform on par with or better than the original IDF model with 8 flows per level (see updated results in Table 1 of the paper). This effectively reduces the number of parameters and time required for a forward pass through the model by a factor of 2, thus making a significant step towards making the use of IDF-like models for real-world compression tasks practical. We hope that in addition to the improvements that we have already shown in the submitted paper, these new results convince the reviewer that the results are relevant for the research community.
>
> **Other comments and questions: clarity of gradient bias section**
> We are motivated to improve the clarity of this section. If the reviewer could point us to specific things that are unclear or could be improved, it would help us greatly with identifying the directions for improvement.
>
> **Other comments and questions: cost of computing finite element analysis**
> Yes, finite differences are very expensive to compute, and indeed therefore the finite difference computations are computed only for a small model (only 2 coupling layers) and on the toy example 1-bit and 8-bit datasets. This is stated in the second paragraph of page 6 in section 5 and more details on this can be found in Appendix B and D3.

---

> > ### Author Response · Authors · 2020-11-17
> > **Response 2/2 to review by AnonReviewer2**
> >
> > **Other comments and questions: Why does inverting the channels help?**
> > The inverting of the permutations of the channels affects how the data is split after each level. At the start of each level, the data is pushed through a space-to-depth transformation, effectively leading to a stack of lower resolution versions of the R, G and B channels. At the end of each level, the resulting stack is split in half along the channel dimension, with one half serving as input to the next level, and the other half no longer transformed by any flows. The distribution of this second half is modeled as conditioned on the first half of the stack. Furthermore, each level contains alternating channel permutations and coupling layers. Therefore, without inverting the channel permutations, these permutations influence the way the data is split, which in turn then influences the factorization of the probability of the image across pixels/dimensions. If we invert the permutations, this influence on the split after each level is removed (not the splitting inside each coupling layer), and we are ensured that splitting is done according to the channels ordered by the space-to-depth transformations. We have clarified this in Section 6 and in a more detailed way in Section D2 in the appendix and Figure 10 for a visualization.
> >
> > **Other comments and questions: evaluating with an exponential moving average**
> > The exponential moving average here refers to the use of an exponential moving average of the model parameters, which is tracked during training. No special exponential moving average is used in GroupNorm. We have clarified this in the updated submission.

---

> > > ### Comment · AnonReviewer2 · 2020-11-19
> > > **Thanks for response and updated review**
> > >
> > > I appreciated the author response and have updated my score accordingly.

---

### Official Review · AnonReviewer3 · 2020-10-28

**Rating:** 7
**Confidence:** 4

**Review:**

This paper aims to analyze and improve IDFs for lossless compression. The authors claim is mainly three following points:
1. IDFs treat the random variable as integers with a countably infinite number of classes, so it does not have a limited factorization capacity.
2. Hoogeboom et al. (2019) demonstrated that the performance of IDFs can deteriorate when the number of coupling layers is too large, gradient bias induced by the straight-through estimator (STE) was suggested to be the cause. In this paper, authors claim that gradient bias due to the STE is less of a problem, and it highly depends on architectural choice of coupling layer.
3. Some architectural changes (invert the channel permutations, rezero trick, group normalization) can get better results.

Pros:
- It is interesting that several types of quantizing functions are evaluated (Fig. 3) and the results shows that STE is less of a problem.
- Architectural changes seems make better results than conventional IDFs (Table 1).
- Authors shows theoretical background of IDFs and theoretical analysis and characteristics are well written.

Cons:
- Hoogeboom et al. (2019) shows adding more flow layers than 16 coupling layers per level hurts performance as is depicted (Figure 5 at Hoogeboom et al. (2019)). But in this paper, Fig. 3 shows only 8 coupling layers experimental results is shown, so I think it is little weak as a rationale for above claim 2..
- In contrast to the author's finding (It can be deeper depending on the architecture) in Chapter 5, IDF++ optimization in Chapter 6 was seems to relatively close to detailed tuning, and relevance is a little weak. It seems more convincing if you examine the effect of deepening the coupling layers with IDF++.

---

> ### Author Response · Authors · 2020-11-17
> **Response to review by AnonReviewer3**
>
> We thank the reviewer for their comments and suggestions. We appreciate that the reviewer lists as positive points the analysis of different gradient estimators, the improved results due to architecture changes and the quality of writing of the theoretical analysis. In the remainder we focus on the two cons raised by the reviewer.
>
> **Performance as a function of flows per level and comparison with Hoogeboom et al.**
> The plot in Fig 5 of the paper by Hoogeboom et al. corresponds to a model with very shallow ConvNets as coupling layers for up to 32 flows per level. For this particular choice of coupling layers it shows that the performance indeed deteriorates after 16 flows per level. However, this architecture is not comparable to the architecture that was used for the best results by Hoogeboom et al. The best reported results correspond to models with DenseNet coupling layers. Because of the much larger capacity of these coupling layers, these larger DenseNet models only use 8 flows per level.
>
> This is also reflected by the performance in Fig 5 of Hoogeboom et al. It is far from the best performance reached with their larger DenseNets: in table 3 of Hoogeboom et al. the best result is 3.32 bits-per-dim for CIFAR-10 versus $\pm 3.64$ bpd for the best discrete ConvNet model in their Figure 5.
>
> Our goal is to improve upon the best model of Hoogeboom et al., therefore we plot the performance for architectures similar to their best model in Fig. 3. In our plot in Fig 3 we include the DenseNet architecture of the best IDF model, and ResNets and ConvNets of comparable size (thus much larger/deeper than in Fig 5 of Hoogeboom et al.). The trend for the ConvNet model is the same as in the original figure of Hoogeboom et al., but with these larger capacity coupling layers the performance deteriorates more quickly as a function of the flows per level. Note though that the best discrete ConvNet result in our paper (Fig. 3) for an unconverged model is already slightly better than the best discrete ConvNet result in Fig 5 of Hoogeboom et al. ($3.58$ bpd versus $\pm 3.64$).
>
> **Additional experiments**
> Taking into account the suggestions by the reviewer, we have performed the following additional experiments.
> * IDF with the ConvNet, ResNet and DenseNet architectures up to 10 flows per level evaluated on CIFAR10. Going for more than 10 flows per level with these architectures requires a lot of memory and takes longer. Figure 3 in the submission has been updated with these results.
> * Per the reviewer’s suggestion, we also included the performance of the DenseNet++ architecture (used in IDF++) for 2, 4, 8, 10 flows per level for CIFAR10 in the updated Figure 3. These results show that the proposed DenseNet++ architecture has more benefits than simply improved performance. The more efficient learning also allows for smaller models to perform as well as the original IDF model. For the IDF++ (DenseNet++) model the performance of _both_ the continuous and discrete model decreases due to overfitting for 10 flows per level. We will add a plot to the appendix in the next few days that shows that these models are overfitting.
> * The IDF++ model with 4 flows per level was run until convergence with the full training set used for training, for CIFAR-10, ImageNet-32 and ImageNet-64. The results can be found in the updated Table 1. These new results demonstrate that IDF++ can perform on par or better than the baseline IDF model with only half the number of flows per level, reducing the number of parameters  and increasing the runtime with an approximate factor of 2.
>
> We thank the reviewer for suggesting the additional experiments, and we hope that these new results and the clarifications will convince the reviewer that Fig 3 depicts the most relevant information for analyzing the influence of flow depth for IDF-type models, and that the analysis of IDF++ as a function of flows-per-level clearly shows that the proposed architecture modifications have beneficial consequences in terms of efficiency.

---

> > ### Author Response · Authors · 2020-11-19
> > **appendix updated**
> >
> > As promised in our previous response, we have updated the appendix of the submission with a plot that shows that an IDF++ model and its continuous counterpart start to overfit for 10 flows per level.

---

> > > ### Comment · AnonReviewer3 · 2020-11-20
> > > **Thanks for the update**
> > >
> > > I really appreciate the authors' update and careful explanation.
> > > The addition of DenseNet++ to Figure 3 makes the advantages of the authors' method more clear.
> > > I think such improvements in compression techniques, even small ones, are very important.
> > > Now that my concerns have been cleared up, I'll give it a higher rating.

---

### Author Response · Authors · 2020-11-17
**Rebuttal overview**

We thank all reviewers for their feedback. We will respond to each reviewer separately so we can best address each comment. Based on the reviewer’s comments we have made several adjustments to the text, which we indicate in the relevant responses to each reviewer. As per the reviewer’s suggestions we have also performed additional experiments, which we list here for an overview:

* As per the suggestion of AnonReviewer3, we studied the performance of an IDF++ model as a function of flows per level. The results show that IDF++ models (with the proposed alterations of the DenseNet architecture, indicated with DenseNet++) outperform models with different architectures for flows per level ranging from 2 to 10. See the updated Figure 3.
* Fig 3 now contains the performance of IDF with different coupling layer architectures for 2, 4, 8 and 10 flows per level, as opposed to the previous version that only contained up to 8 flows per level.
* We show that a converged IDF++ model with 4 flows per level can perform on par with or better than an IDF model with 8 flows per level on CIFAR-10, ImageNet-32 and ImageNet-64. This effectively reduces the number of parameters by a factor of 2 and also reduces the run time of the model. See the updated results in Table 1.

In the remainder of the responses, when we refer to the works by Hoogeboom et al. and Papamakarios et al., we are referring specifically to:

[1] Emiel Hoogeboom et al. Emerging Convolutions for Generative Normalizing Flows. In Proceedings of the 36th International Conference on Machine Learning, 2019.
[2] George Papamakarios et al. Normalizing Flows for Probabilistic Modeling and Inference. arXiv:1912.02762, 2019.

---

### Decision · Program_Chairs · 2021-01-07
**Final Decision**

**Decision:**

Accept (Poster)

**Comment:**

The reviewers of this paper unanimously agreed that this paper adds an interesting theoretical and practical discussion to discrete flows. The paper has improved from the first version to the final one, in which the comments and suggestions by the reviewers have been followed.

The paper is still incremental with respect to the previous paper and the reviewers all recommended a poster presentation.